# Impact of KIR-HLA Genotype on Natural-Killer-Cell-Based Immunotherapy for Preventing Hepatocellular Carcinoma after Living-Donor Liver Transplantation

**DOI:** 10.3390/cancers16030533

**Published:** 2024-01-26

**Authors:** Naoki Tanimine, Masahiro Ohira, Emi Kurita, Ryosuke Nakano, Hiroshi Sakai, Hiroyuki Tahara, Kentaro Ide, Tsuyoshi Kobayashi, Yuka Tanaka, Hideki Ohdan

**Affiliations:** 1Department of Gastroenterological and Transplantation Surgery, Graduates School of Biomedical and Health Sciences, Hiroshima University, 1-2-3 Kasumi, Minami-ku, Hiroshima 734-8551, Hiroshima, Japan; tanimine@hiroshima-u.ac.jp (N.T.); mohira@hiroshima-u.ac.jp (M.O.); rynk@hiroshima-u.ac.jp (R.N.); sakaih@hiroshima-u.ac.jp (H.S.); htahara@hiroshima-u.ac.jp (H.T.); ideken@hiroshima-u.ac.jp (K.I.); tsukoba@hiroshima-u.ac.jp (T.K.); yukasan@hiroshima-u.ac.jp (Y.T.); 2Department of Surgery, Kure Medical Center, Chugoku Cancer Center, 3-1 Aoyama-cho, Kure 737-0023, Hiroshima, Japan; 3Division of Blood Transfusion Services, Hiroshima University Hospital, 1-2-3 Kasumi, Minami-ku, Hiroshima 734-8551, Hiroshima, Japan; kurita@hiroshima-u.ac.jp

**Keywords:** hepatocellular carcinoma, natural killer cell, human leukocyte antigen, killer immunoglobulin-like receptor, liver transplantation, immunotherapy, recurrence, adjuvant therapy

## Abstract

**Simple Summary:**

Natural killer (NK) cells have immunosurveillance potential in hepatocellular carcinoma. Human leukocyte antigen (HLA) self-recognition through killer immunoglobulin-like receptor (KIR), a process termed “licensing”, raises the NK cell capacity. The polymorphic KIR-HLA genotype revealed that genetically vulnerable liver transplant recipients with a poorly licensed NK genotype have an improved prognosis by immunotherapy with donor-liver-derived NK cells. Thus, the combination of recipient and donor KIR-HLA genotypes is worthy of attention for further investigation, especially considering the clinical application of NK-cell-based immunotherapy.

**Abstract:**

Natural killer (NK) cells have immunosurveillance potential in hepatocellular carcinoma (HCC). We performed adaptive immunotherapy using donor-liver-derived natural killer (NK) cells after living-donor liver transplantation (LDLT) to prevent HCC recurrence. Dominant inhibitory signals tightly regulate NK cell activity via human leukocyte antigen (HLA)-specific inhibitory receptors, such as killer immunoglobulin-like receptors (KIRs). The functional recognition of HLA through KIR raises the NK cell capacity, which is a process termed “licensing.” Here, we investigated the effect of polymorphic KIR-HLA genotypes on the efficacy of NK-cell-based immunotherapy after LDLT. Seventy-seven Japanese recipients with HCC who underwent LDLT and their corresponding donors between 1996 and 2016 were enrolled in this study. The median follow-up period was 8.3 years. The HCC recurrence risk was stratified using radiological and pathological assessments according to the Milan criteria. Of the 77 recipients, 38 received immunotherapy. Immunotherapy improves early post-transplantation survival and lowers the recurrence rate in the intermediate-risk recipients. We analyzed the genotypes of five inhibitory KIRs and HLA using sequence-specific polymorphism-based typing. The polymorphic KIR-HLA genotype revealed that genetically vulnerable liver transplant recipients with a poorly licensed NK genotype have an improved prognosis by immunotherapy with donor-liver-derived NK cells. Thus, the combination of recipient and donor KIR-HLA genotypes is worthy of attention for further investigation, especially considering the clinical application of NK-cell-based immunotherapy.

## 1. Introduction

Liver transplantation (LT) is the only curative therapy for hepatocellular carcinoma (HCC) in patients with liver failure. Ensuring HCC elimination by pre-transplant risk assessment or adjuvant therapy can provide a long-term prognosis for cirrhotic patients with HCC. Given the poor prognosis of post-liver transplant HCC recurrence under the immunosuppressive condition, preventing HCC recurrence is crucial. Recurrence after removing the entire native liver is likely caused by systemically spread tumor cells. Although the emerging systemic chemotherapy regimens have shown great therapeutic effects on some advanced HCC, they are not allowed to be applied in the early period after liver transplantation because of their negative effect on wound healing and the potential to induce rejection. Randomized control studies with adoptive transferred autologous ex vivo cytokine-inducing killer cells for HCC patients receiving locoregional therapy have shown their feasibility and the impact on lowering the recurrence rate or prolonging the time to recurrence [1,2,3,4]. Adoptive immunotherapy is a promising strategy for preventing HCC recurrence by scanning the whole body, unless it induces an immune response against allografts. Based on the remarkable cytolytic activity of natural killer (NK) cells in the healthy liver, especially against HCC [5,6], we applied adjuvant immunotherapy with donor-liver-derived natural killer (NK) cells for HCC patients with liver failure [7,8]. The use of allogeneic NK cells for cancer treatment has been studied as a graft-versus-leukemia effect in the field of stem cell transplantation [9]. Recent studies have shown that allogeneic NK cells have a better prognosis in patients with advanced HCC [10,11]. NK cells are tightly regulated by dominant inhibitory signals by human leukocyte antigen (HLA)-specific inhibitory receptors such as killer immunoglobulin-like receptors (KIRs), which provide the “missing-self” mechanism on tumor elimination [12]. The functional recognition of HLA through KIR raises the NK cell capacity, a process termed “licensing” [13,14]. KIRs and HLA ligands are polymorphic and diverse immune protection by the multiplicity of functional compound KIR-HLA genotypes through NK cell activity has been observed after curative HCC resection [15]. The self-protective mechanism of NK cells can regulate NK activity in an allogeneic setting. Here, we investigated the effect of the KIR-HLA genotype on the efficacy of NK-cell-based immunotherapy after living-donor liver transplantation.

## 2. Material and Methods

### 2.1. Patients and Outcomes

In this study, 77 Japanese patients with HCC who underwent living-donor liver transplantation (LDLT) at the Hiroshima University between 2006 and 2016 were enrolled based on the following inclusion criteria: preoperative diagnosis of HCC and availability of both recipients and donors for KIR genotyping. Of the 77 patients, 38 received adjuvant immunotherapy with donor-liver-derived NK-cell-enriched lymphocytes [7]. This study was approved by the Hiroshima University Research Ethics Committee (Hi-77), and written informed consent was obtained from all patients in accordance with the Declaration of Helsinki. We administered a preconditioning regimen comprising rituximab, tacrolimus, and mycophenolate mofetil combined with double-filtration plasmapheresis to ABO-incompatible patients. After LT, we employed an immunosuppressive protocol, including methylprednisolone and a calcineurin inhibitor (CNI), considering a CNI-sparing regimen with mycophenolate mofetil for patients with chronic kidney disease. Tacrolimus was selected as the standard CNI, but cyclosporin A was selected for patients with HCV infection based on the suppressive capacity for HCV replication [16]. The same immunosuppression protocol was used for recipients with and without immunotherapy. Methylprednisolone was gradually tapered within one year in the absence of an immunological event. Immunosuppression was adjusted depending on whether clinical or biopsy-proven rejection occurred.

Clinicopathological and follow-up data were collected five years after liver transplantation. After liver transplantation, patients were followed-up using ultrasound sonography, contrast-enhanced computed tomography, or magnetic resonance imaging combined with evaluation of serum alpha-fetoprotein (AFP) and des-gamma carboxyprothrombin levels at 3-month intervals for up to 3 years. Thereafter, follow-ups were performed at 6-month intervals for up to 5 years. HCC recurrence was defined as the appearance of a new focal hepatic or extrahepatic lesion, including in the lungs and bone, with typical characteristics. Any episode of bloodstream infection (BSI) that developed within 90 days after surgery, documented as the first infection episode, was corrected. 

### 2.2. Immunotherapy with Donor-Liver-Derived NK Cells

To prevent the recurrence of HCC and HCV infection, adoptive immunotherapy with donor-liver-derived NK cells was approved by the Clinical Institutional Ethical Review Board of Hiroshima University (Rin 414-1 and 40019) and registered with the UMIN (000012162 and 000000538) as a phase 1 clinical trial. The exclusion criterion for the HCC study (UMIN00012162) was absence of HCC. The exclusion criteria for the HCV study (UMIN00000538) were uncontrolled infection, fulminant hepatitis, ABO-incompatible transplantation, or repeat transplantation. Collectively, any HCC status, except for the existence of distal metastasis or macrovascular invasion, was allowed for enrollment in these studies based on the expanded indication criteria for the living-donor setting. Some patients in the same period were not offered the trial because of the limited capacity of the cell processing center, or did not provide consent for enrolment. The adoptive immunotherapy protocol has been previously described [7]. Briefly, after the donor hepatectomy, ex vivo perfusion of the liver allograft was performed via the portal vein. Liver-allograft–derived lymphocytes were isolated by gradient centrifugation. Liver lymphocytes were cultured with human recombinant IL-2 (100 Japanese reference units/mL [JRU/mL]) for 3 days. One day before the infusion, 1 μg/mL of OKT3 (Janssen Pharmaceutical K. K. Tokyo, Japan) or anti-CD3 monoclonal antibody (Miltenyi Biotec K.K. Tokyo, Japan) was added to opsonize the CD3+ fraction. Three days after LT, activated liver-NK-cell-enriched lymphocytes were administered via venous circulation. Cell viability was assessed using a dye-exclusion test, and the cells were checked twice for possible contamination with bacteria, fungi, and endotoxins.

### 2.3. KIR and HLA Genotyping

Genetic analyses were performed as previously described [15]. Genomic DNA was extracted from peripheral blood mononuclear cells derived from patients by using a QIA cube (Qiagen, Hilden, Germany). KIR allele genotyping for KIR2DL1/2DL2/2DL3/3DL1/3DL2 was performed by sequencing KIR transcripts and detected using the reverse sequence-specific polymorphism-polymerase chain reaction (SSP-PCR)-Luminex typing method using a KIR genotyping SSO kit (One Lambda, Canoga Park, CA, USA). The presence of the HLA ligand for KIR was determined based on the ligand specificities for five inhibitory KIRs: KIR2DL1 for the HLA-C Lys80 (C2) group of alleles, KIR2DL2 and KIR2DL3 for the HLA-C Asn80 (C1) group, KIR3DL1 for the Bw4 group of HLA-B (and some A) alleles, and KIR3DL2 for the HLA-A3/11 alleles [17].

### 2.4. Flow Cytometry Analysis

The following antibodies were used: APC-H7-CD3 (HIT3a), BV510-CD56, FITC CD158a (HP-3E4), PECy7-CD158b (CH-L), BV421-CD158e (NKB1), and PE-CD158k (p140), purchased from BD Biosciences (San Jose, CA, USA). Dead cells were excluded from the analysis using DAPI (Thermo Fisher, Tokyo, Japan) staining. Data were collected using an LSR Fortessa X-20 or FACS Canto II and analyzed using FlowJo v. 10 (Tree Star, Ashland, OR, USA).

### 2.5. Statistical Analysis

Categorical and continuous variables were compared using the chi-square and Wilcoxon tests, respectively. The cumulative risk of recurrence (CRR) and overall survival (OS) were estimated and compared using Kaplan–Meier and log-rank statistics. Statistical analyses were performed using JMP16 for Windows (SAS Institute, Cary, NC, USA). *p* values < 0.05 were considered significant.

## 3. Results

### 3.1. Donor-Liver-Derived NK Immunotherapy Improved the Survival of the Recipients with HCC

The median follow-up period was 8.3 years. Sixty-six and eleven recipients were categorized radiologically as within and beyond the Milan criteria (MC, single tumor ≤ 5 cm, or up to 3 tumors ≤ 3 cm, without extrahepatic metastases or vascular invasion) by preoperative radiological assessment [18]. By post-transplant histological examination, we defined 23 of 66 radiologically within MC recipients as pathologically beyond MC, in which the histological tumor number or diameter was beyond MC or the presence of microvascular invasion (Figure 1A). The categorized group based on radiological and pathological assessments (pathologically within MC, beyond MC, and radiologically beyond MC) stratified the risk of HCC recurrence after transplantation (5-years CRR; 0%, 25.2%, and 70.4%, respectively; Figure 1B, Appendix A). Of the 77 recipients, 38 recipients received adjuvant immunotherapy with donor-liver-derived NK cells. Recipients who received immunotherapy showed significantly better survival in the early post-transplantation period (1-year OS with or without immunotherapy, 97.4% vs. 81.8%, *p* = 0.027; Figure 2A). The frequency of BSI was significantly lower in recipients who received immunotherapy than in those who did not (2.6% vs. 18.0%, *p* = 0.020). The CRR were comparable, possibly because of the advanced tumor status in the immunotherapy group (Figure 2B, Appendix A). Recipients receiving immunotherapy showed favorable CRR in the subgroup analysis of intermediate recurrence risk, pathologically beyond MC recipients, although the high recurrence risk, radiologically beyond MC recipients, appeared to be far more advanced to benefit from immunotherapy (Figure 2C,D). 

### 3.2. Status of KIR-HLA Genotype in Living-Donor Liver Transplant Recipients and Donors

Functional compound KIR-HLA genotypes (KIR2DL1-C2, KIR2DL2-C1, KIR2DL3-C1, KIR3DL1-Bw4, and KIR3DL2-HLA A3/11), which intrinsically license NK cells, were found equally in recipients and donors (Table 1). All recipients and donors had between one and four of the five functional compound KIR-HLA genotypes, and their frequency was comparable to that in a previous report from an HCC cohort without liver failure. 

Focusing on KIR-HLA genotype multiplicity, a high NK-licensing genotype, defined as possession of more than three functional compound genotypes, which have been proven to have an impact on HCC recurrence after primary hepatectomy [15], was found in 26.0% and 28.6% of recipients and donors, respectively (Table 1). No protective effect of the recipient’s highly NK-licensed genotype on OS or CRR in the overall cohort analysis was observed (Figure 3A,B; subgroup background characteristics are shown in Appendix A). Subgroup analysis based on receiving immunotherapy demonstrated that recipients with a poorly NK-licensed genotype benefited from immunotherapy with significantly better survival compared to those with the same genotype without immunotherapy (Figure 3C). The recipient-licensed NK genotype did not show an effect on immunotherapy in CRR analysis (Figure 3D), and, unpredictably, the recipient highly NK-licensed genotype was associated with higher CRR in the subgroup analysis of pathologically beyond MC recipients (*p* = 0.048). The higher CRR was likely biased by tumor characteristics, which is representative of significantly high APF levels in the highly licensed group (median AFP level 19.8 vs. 1747 ng/mL, *p* = 0.027; recipient characteristics in Appendix A).

### 3.3. Effect of KIR-HLA Genotype Combination on Allogeneic NK Cell Immunotherapy for HCC Recurrence after Liver Transplantation

In the field of hematopoietic stem cell transplantation, donor NK cell licensing and an increasing number of mismatches between donor inhibitory KIRs and cognate HLA ligands in recipients (receptor–ligand mismatch) have shown favorable outcomes in preventing AML recurrence [9,19,20,21]. Liver NK cells from donors with a highly licensed NK genotype were abundant among the multiple KIR-expressing NK cells (Appendix A) and preserved KIR expression after IL-2 activation (Appendix A). Based on this observation, we investigated whether the donor-NK-cell-licensed genotype and receptor–ligand mismatch status affect the efficacy of immunotherapy in HCC recurrence. In high recurrence risk, radiologically beyond MC recipients, the donor-NK-licensed genotype and receptor–ligand mismatch status of the genotype did not show a recognizable effect on recurrence. However, in intermediate recurrence risk, the pathologically beyond MC recipients, recipients with immunotherapy from highly NK-licensed genotype donor did not experience a recurrence although all the recipients without immunotherapy exhibited a recurrence (Figure 4A; individual tumor characteristics are presented in Appendix A). Additionally, a recipient with three receptor–ligand mismatches did not experience recurrence with immunotherapy. The CRR curve was not significant but was lower in recipients with immunotherapy in comparison to those without immunotherapy with the same number of receptor–ligand mismatches, and the recurrence rate gradually decreased according to the number of receptor–ligand mismatches in recipients with immunotherapy (5yrs-CRR of 1 mismatch, 2 mismatches, and 3 mismatches: 25.0%, 11.1%, and 0%, respectively; Figure 4B). 

## 4. Discussion

Protection against cancer recurrence and infection involves both innate and acquired immunity. Under conditions of immunosuppressive therapy for preventing rejection after solid organ transplantation, acquired immunity consisting of T and B cells is preferentially suppressed by current immunosuppressive regimens. After transplantation, when acquired immunity is intentionally suppressed, it is reasonable to enhance the immune surveillance of innate immunity by adaptive immunotherapy with activated NK cells. Since liver NK cells uniquely possess antiviral and anti-HCC activity [5,6], we used activated donor-liver-derived NK cells to prevent HCV and HCC recurrence after liver transplantation. Here, we observed the survival benefit of NK-cell-based immunotherapy for early post-liver LDLT in HCC patients, regardless of the HCC characteristics. Severe infections, such as BSI, have been reported to be associated with poor prognosis after liver transplantation [22]. Recipients receiving immunotherapy exhibited a better prognosis and lower incidence of BSI. The survival benefit of KIR-HLA genotypes in a liver transplantation cohort has been investigated and remained controversial [23,24,25]. Recent analyses have reported the immunological impact of a combination of recipient KIR and the presence of a corresponding donor ligand on rejection due to the alloreactivity of recipient NK cells on allografts [26,27]. Here, we observed that immunotherapy preferentially rescued recipients with a poorly licensed NK genotype, who are likely to have “genetic vulnerability” in innate immunosurveillance. The effect of the recipient-licensed NK cell genotype on HCC recurrence remains unclear because the aggressive HCC status of the highly licensed NK cell recipients can compromise the results and is difficult to adjust for in our small cohort. However, recipients with a poorly licensed NK genotype could be good candidates for immunotherapy with donor-liver-derived NK cells because of their vulnerability to the immunosurveillance of infection, especially under unusual biophylactic conditions in which acquired immunity should be suppressed.

Although preoperative assessment using standardized criteria such as MC stratifies the risk of HCC recurrence well, the actual tumor status is often recognized by pathological assessment after transplantation [18,28]. A long-term prognosis is not expected once recurrence occurs with immunosuppressive therapy [29,30]. However, to our knowledge, currently, there is no effective option for preventing HCC recurrence after liver transplantation in current practice [31,32,33]. We observed the effect of immunotherapy with donor-liver-derived NK cells in preventing HCC recurrence after LDLT in recipients with intermediate HCC recurrence risk. To enhance the protective effect, the donor-licensed NK genotype can rationally predict the alloreactivity of adoptive NK cell therapy through the education process to become fully functional [19]. Although the expression of KIRs can be modified by ex vivo treatment [34], IL-2-activated NK cells preserved the expression of inhibitory KIRs. KIR-expressing allogeneic NK cells are expected to be reactive when the recipient HLA ligand mismatches with donors possessing an inhibitory receptor (receptor-ligand mismatch model) [9,20,21]. We observed trends of protection from HCC recurrence in recipients with an intermediate recurrence risk receiving immunotherapy from donors with a highly NK-licensed genotype, but not in recipients with a high recurrence risk of HCC. Collectively, recipients with poorly licensed NK cell genotypes are good candidates for immunotherapy, achieving better survival rates during the early postoperative period. Consistent with the hypothesis, the patients pathologically beyond the Milan criteria with immunotherapy from the donor with a highly licensing genotype had a clinical presentation that prevents HCC recurrence despite the limited cases. The effect of the receptor–ligand mismatch on immunotherapy remains unknown for our dataset. Current data suggest that a favorable donor selected based on the KIR-HLA genotype for immunotherapy with donor-liver-derived NK cells potentially eliminates the intermediate risk of HCC recurrence after liver transplantation. Selecting organ donors based on genotyping is difficult in living-donor settings; it may be possible to consider diseased donor settings in the future to improve the prognosis of HCC recipients. Furthermore, to eliminate advanced HCC, additional strategies, such as ex vivo manipulation by chimeric antigen receptors or gene editing, are required during the expansion or differentiation induction of functional NK cells. 

When interpreting the results of the current analysis, it is important to consider the limitations of the dataset. The small sample size was insufficient to determine the clinical management. However, we can suggest the favorable effects of the KIR-HLA genotype combination to improve the effect of NK-cell-based immunotherapy for survival and HCC recurrence to conduct a nationwide, large-scale study. 

## 5. Conclusions

We observed that immunotherapy with donor-liver-derived NK cells showed a survival benefit in the early post-LDLT period and prevented HCC recurrence in recipients with an intermediate recurrence risk. The KIR-HLA genotype revealed that genetically vulnerable recipients with poorly licensed NK cell genotypes can be rescued by immunotherapy. The combination of recipient and donor KIR-HLA genotypes is worthy of attention as an option to enhance the therapeutic capacity of NK-cell-based immunotherapy in HCC recipients receiving liver transplantation.

## Figures and Tables

**Figure 1 cancers-16-00533-f001:**
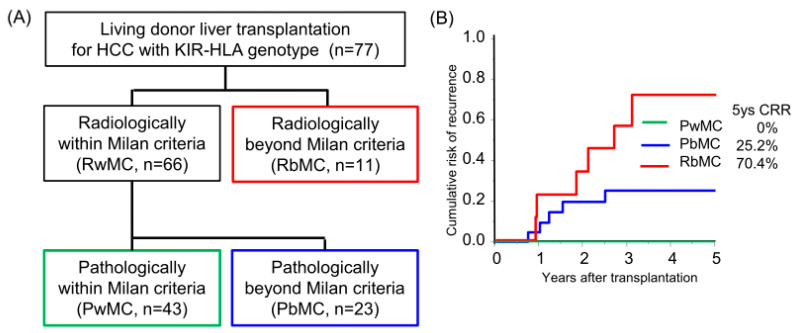
Risk category of hepatocellular carcinoma recurrence after living-donor liver transplantation by Milan criteria. (**A**) The flow diagram shows the radiological and pathological assessments using the Milan criteria (single tumor ≤ 5 cm or up to 3 tumors ≤ 3 cm, without extrahepatic metastases or vascular invasion). Pathological assessment was performed to evaluate the number and diameter of viable lesions and the presence of microvascular invasion. (**B**) Cumulative risk of recurrence (CRR) in patients with HCC who underwent living-donor liver transplantation according to radiological and pathological assessments. Patients pathologically within the Milan criteria did not experience recurrence.

**Figure 2 cancers-16-00533-f002:**
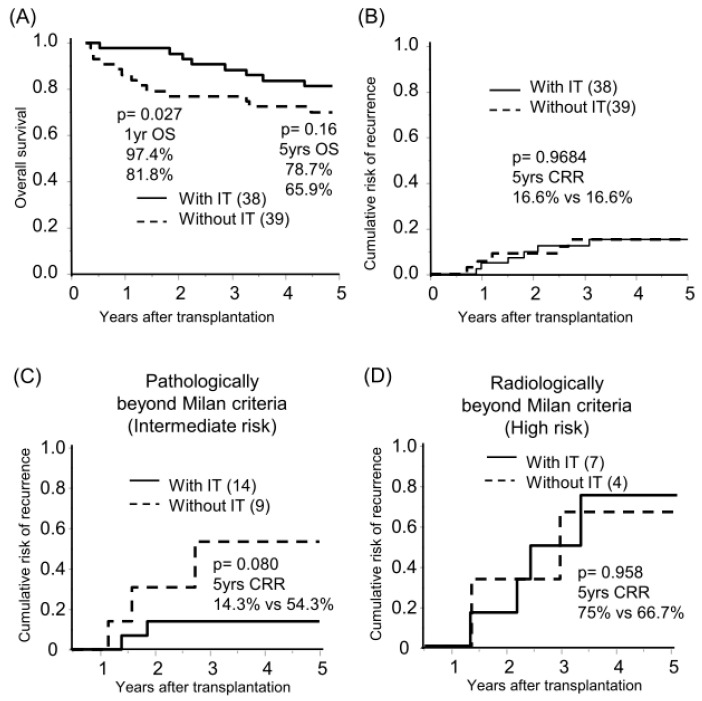
Benefit of immunotherapy with donor liver-derived NK cells on survival and HCC recurrence. Kaplan–Meier analyses of the 5-year overall survival (**A**) and cumulative risk of recurrence (**B**–**D**) were performed according to immunotherapy (with IT/without IT). Analyses were performed for all patients (*n* = 77, (**A**,**B**)), those pathologically beyond the Milan criteria (*n* = 23, (**C**)), and those radiologically beyond the Milan criteria (*n* = 11, (**D**)). Statistical analyses were performed using the log-rank test.

**Figure 3 cancers-16-00533-f003:**
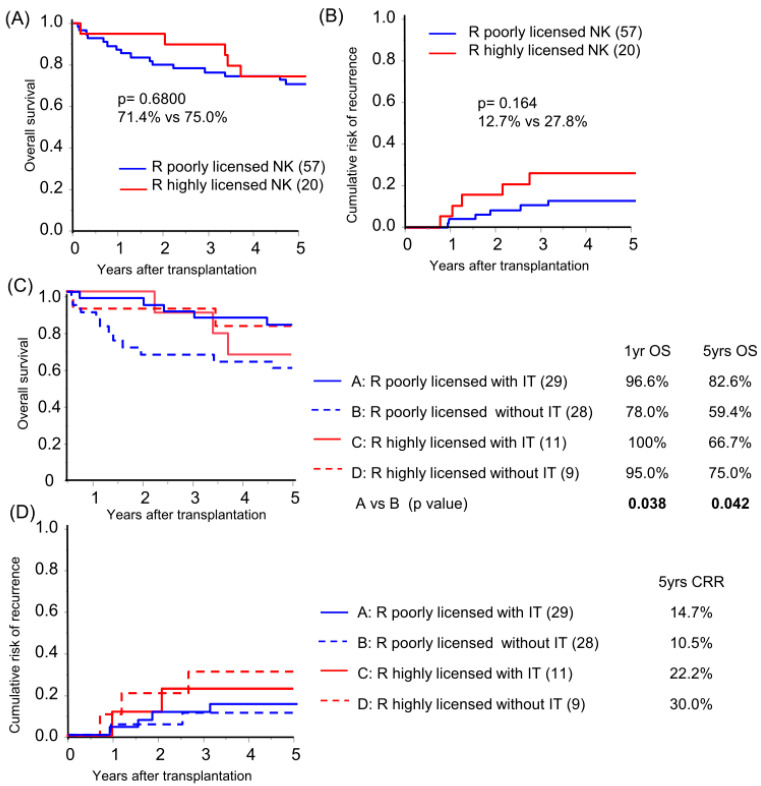
Recipient NK-licensed genotype was associated with susceptibility for immunotherapy with donor-liver-derived NK cells. Kaplan–Meier analyses of the 5-year overall survival (**A**,**C**) and cumulative risk of recurrence (**B**,**D**) were performed for patients with >3 functional compound KIR-HLA genotypes (R highly licensed NK) and for patients with one or two compound genotypes (R-poorly licensed NK). Subgroup analyses were performed according to immunotherapy (IT/without IT; (**C**,**D**)). Statistical analyses were performed using the log-rank test.

**Figure 4 cancers-16-00533-f004:**
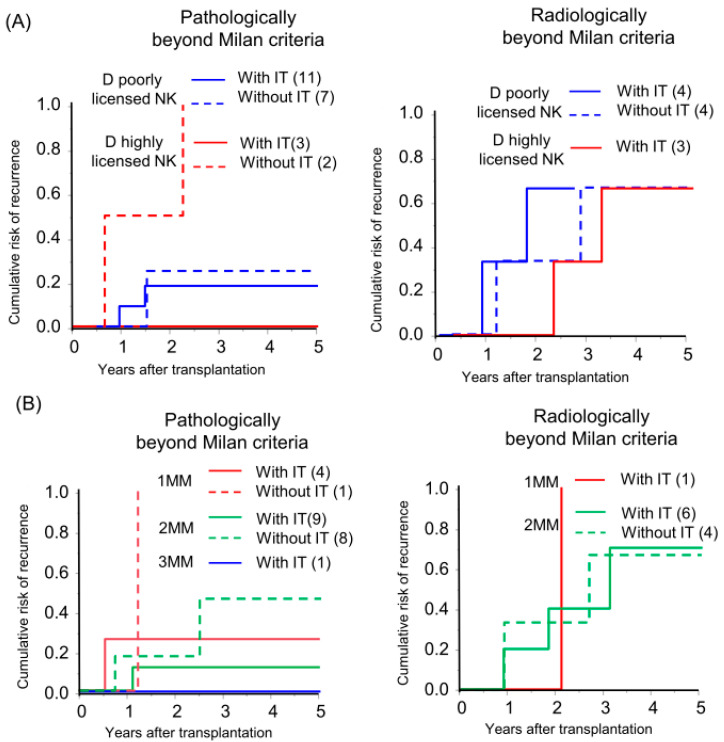
Donor-NK-licensed genotype and receptor–ligand mismatch status were associated with the prevention of HCC recurrence by immunotherapy with donor-liver-derived NK cells. Kaplan–Meier analyses of the 5-year cumulative risk of recurrence were performed for (**A**) patients with a donor who had at least three functional compound KIR-HLA genotypes (D highly licensed NK) and patients with a donor who had one or two compound genotypes (D poorly licensed NK), and (**B**) the number of receptor (donor)–ligand (recipient) mismatches (MM) with subgrouping according to receiving immunotherapy (with IT/without IT).

**Table 1 cancers-16-00533-t001:** Summary of KIR and HLA genotyping analyses.

	Recipient (*n* = 77)		Donor (*n* = 77)	
Functional KIR-HLA compound	Positive	%	Positive	%
KIR2DL1-C2	11	14.3%	16	20.8%
KIR2DL2-C1	9	11.7%	9	11.7%
KIR2DL3-C1	77	100.0%	77	100.0%
KIR3DL1-Bw4	57	74.0%	52	67.5%
KIR3DL2-A3/11	17	22.1%	21	27.3%
Highly licensed NK genotype (≥3 compounds)	20	26.0%	22	28.6%

## Data Availability

The data presented in this study are available on request from the corresponding author (accurately indicate status).

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
