# Peer review of "Impact of KIR-HLA Genotype on Natural-Killer-Cell-Based Immunotherapy for Preventing Hepatocellular Carcinoma after Living-Donor Liver Transplantation"

_cancers, 2024, doi:10.3390/cancers16030533_

Round 1

Reviewer 1 Report

Comments and Suggestions for Authors

The manuscript by Tanimine et al. explores the use of donor liver-derived natural killer cells in adaptive immunotherapy after living-donor liver transplantation to prevent hepatocellular carcinoma recurrence. It focuses on the impact of polymorphic Killer Immunoglobulin-like Receptor and Human Leukocyte Antigen genotypes on NK cell-based immunotherapy efficacy. The research, involving 77 recipients with HCC who underwent LDLT, indicates that genetically vulnerable liver transplant recipients with poorly licensed NK genotypes show improved prognosis through immunotherapy. The study suggests considering recipient and donor KIR-HLA genotypes in NK cell-based immunotherapy clinical deployment. Some revisions are suggested.

1. The introduction is not adequate, a comprehensive rational as well as other existing preclinical studies and clinical trials should be provided on the use of NK cells in preventing HCC recurrence after LT.

2. There is a lack of information, such as the exclusion criteria for the treatment of NK cells. What is the contraindication for such treatment?

3. The authors mainly used Milan criteria as well as KIR and HLA genotyping to stratify the patients. It would be ideal to summarize which group of patients would benefit the most with NK cell treatment. 

4. What is the immunosuppressive protocol for HCC patients in authors’ center, is the protocol differ between patients with or without NK cell therapy?

Comments on the Quality of English Language

minor

Reviewer 2 Report

Comments and Suggestions for Authors

The study is use of activated donor liver‐derived NK cells to prevent HCC recurrence after liver transplantation.

The authors have concluded that The polymorphic KIR‐HLA genotype revealed that genetically vulnerable liver transplant recipients with a poorly licensed NK genotype have improved prognosis. And they have added that combination of recipient and donor KIR‐HLA genotypes is worthy of attention.

The manuscript is well designed and well written.

This study may be beneficial for preventing recurrence in HCC after liver transplantation.

There are a few comments on this manuscript.

1. Genetic analysis is difficult clinically. Do authors think that this analysis should be done routinely clinically? And the patient group who benefit from this analysis is also limited. Please discuss on this point.

2. As discussed, the study has limited number of patients.

Is there a way to increase the number of patients? Large number of the patient will be good to show more solid data.

3. Of the 77 patients, 38 received adjuvant immunotherapy with donor liver‐derived NK cell‐enriched lymphocytes. How the authors have selected this group?

Round 2

Reviewer 1 Report

Comments and Suggestions for Authors

My questions have been address.